# The Higher Education Commitment Challenge: Impacts of Physical and Cultural Dimensions in the First-Year Students' Sense of Belonging

Diana Dias

School of Economic and Organizational Sciences, Lusofona University, 1749-024 Lisboa, Portugal; diana.dias@ulusofona.pt

**Abstract:** The students' perceptions and experiences about the organizational attributes of the higher education institution in which they are enrolled seem to have a strong influence on their integration, sense of belonging, and commitment to their new academic reality. The present paper focuses on the analysis of how first-year students build a sense of belonging and commitment to the higher education institution that welcomes them, focusing on institutional attributes that can act as (positive or negative) catalysts, such as physical and cultural dimensions. However, besides physical and cultural dimensions, it is crucial to consider its synergies with psychological, social, organisational, political, and axiological dimensions that have emerged as critical variables for contextualizing the analysis. The results suggest that the physical dimension nourishes the students' feelings of belonging, namely through the felt need to develop skills to manage their interaction with the spatial dimension of the institution that welcomes them. Moreover, newcomers' self-concept seems to be significantly increased by the feeling that they are now part of a cultural but also social elite. On the other hand, the feeling of integration seems to be supported basically on successful peer relationships. This perceived prestige of the higher education institution where they now belong represents, *a anteriori*, a crucial demand for the career management of the Bourdieu' "heirs," and, *a posteriori*, a real (and sometimes surprising) achievement for first-generation students.

**Keywords:** student experience; student engagement; organisational development; first-year experience; academic development





## 1. Introduction

Tavares [1] characterises Higher Education Institutions (HEIs) as "reflective, dynamic, flexible, resilient, learning realities". Based on their complex nature, the author advises a multifaceted and comprehensive analysis, considering different dimension, such as physical, psychological, social, organisational, political and cultural, and axiological dimensions.

The physical dimension refers to the infrastructures that encompass the buildings, their external and internal configuration, their design and architectural goal, and their layout and material resources. In fact, the design, disposition, and use of school buildings transmit educational and social values; thus, many of the psychological and social problems emerging in the educational community may be prevented, remedied, or even fixed through specific interventions in the physical surroundings [2–5].

The psychological and sociological dimension are also critical perspectives, as schools are only possible and justified by the people as members of the educational community. Thus, the psychological well-being depends on the capacity (and even need) of each student to connect with others. The relational issue is a core variable to any organisational analysis because the relationship conditions, for not only the processes but also of the results from the individual and the collective action.

Concerning the organisational dimension, the governance of HEIs, management strategies, and funding sources are the focus. On the other hand, the political and the cultural

dimensions must also be considered. Ideologies remain the benchmark for the development of policies, shaping the institutional culture, grasped as a set of rules underlying an open system of human resources, where beliefs, expectations, concepts, and resources are compromised.

However, since the complementary nature of the institutional dimensions has already been highlighted, it is now important to highlight the transversal perspective: the axiological dimension. According to Tavares [1], the meaning of the changes carried out in the institutions must be reflected not only in the physical spaces, in their actors, in the curricula, and in all the organization and management systems (scientific, educational, research, or extension), but also in the surrounding community, considered at its different levels: micro, meso, exo, and macro.

Students' perceptions and experiences of the organisational attributes of the higher education institution they are enrolled in exert a strong influence in their integration into the academic universe. The present paper focuses on the analysis of the higher education institutional reality, emphasising the importance of the physical and cultural features of each institution that may act as catalysts of new students' integration. Thus, the physical, psychological and social, organisational, cultural and political, and axiological dimensions emerge as a relevant framework for analysis.

## 2. Literature Review

Student sense of belonging is critical to success in the first year of university, yet evidence about how and why various institutional factors influence engagement remains relatively rare. For many students, the transition from high school to HE is a hard hurdle [6,7]. They must learn how to deal with the new learning environment, build new relationships with peers and faculty, and grow into their new role as HE students [8,9].

Four phases are identified in the transition into the HE process [10]: preparation, encounter, adjustment, and stabilisation. In the preparation phase, students ponder about their course programme choice and choose where to enrol. Upon acceptance, students are confronted with a new learning environment and an academic culture. Through this encounter phase, they may experience some tension between their personal learning beliefs and behaviour and the new learning environment, with its own specific academic culture. This tension impacts the development of their role as an HE student. Students develop their identity as HE students, adopt their perceptions and behaviour regarding the new learning environment, and ideally create a supportive network to feel at home and successfully cope with the demands and opportunities in HE [6,10]. This encounter phase usually takes place during the first weeks at HE. In the adjustment phase, the third phase of the transition process, amendments in attitude and behaviour take place gradually during the first year. Lastly, in the fourth and final phase in the transition process, students experience broadly what kind of behaviour leads to satisfying social and academic outcomes, and their attitudes and behaviour tend to stabilise [10,11].

First year students seem to be mostly concerned about two different but complementary issues: developing a sense of belonging in HE and building relationships with peers and faculty within it [12–15]. A sense of belonging refers to feeling at home at an HEI and that he/she fits in, that he/she is a member of one or more communities there, and that he/she is supported at the HE [16,17]. Developing a positive sense of belonging in HE seems crucial for the decision not to leave when one experiences difficulties in adapting to the new environment [18,19]. Previous studies have shown that students' interactions with peers and faculty are important for their experiences in HE. Such interactions can take place formally or informally, either inside or outside of a classroom setting [7,20].

Berger and Braxton [21] developed a model that focuses on organisational features as variables that influence student integration. According to this model, how students perceive and experience the organisational attributes of the higher education institution (HEI) they are enrolled in is a potential source of influence on their integration into the academic universe. The authors add that, more than and beyond the organisational features

related to the structure of an HEI (such as size, selectivity, and ways of control), measures of organisational behaviour on campus influence how students make their academic integration. In this study, the authors argue that the traits students bring along when they join the institution affect the level of initial and subsequent commitment to it; this commitment, in turn, is positively affected by the level of integration into existing social communities in the HEI. Thus, the higher the level of subsequent commitment to the institution, the more likely the student is to remain in it [21]. Therefore, students' integration in HEIs is potentially influenced by how students experience their organisational attributes. According to the authors, organisational attributes, such as participation in organisational decision making, justice administration of policies, and communication, affect students' decision to stay or leave the institution. In this study, Berger and Braxton [21] include, in the model proposed by Tinto [19,22,23], variables that measure students' perceptions about the organisational attributes and seek to analyse the effects of these variables in students' integration.

## 3. Materials and Methods

The present paper reports a case study focus on first-year students enrolled in a programme of electrical engineering, who had been attending HE for six months, in a prestigious Portuguese university. In Portugal, Engineering studies are a very prestigious scientific field for HE candidates [24] and the access to them is highly competitive.

### 3.1. Participants

The sample was composed of 30 students, corresponding to 17.5% of the population. In terms of sampling procedures, the study sought to access a list of the total number of students enrolled in Electrical Engineering, rated according to their Grade Point Average. Despite the clear option for a qualitative approach, in the first moment, a probabilistic sampling method was elected, using a stratified random sample, to ensure the homogeneity of the sample distribution in terms of access grades, gender, age, and socio-cultural background.

The final sample of analysable data encompassed, therefore, 30 first year student interviews. Interviewees were 25 male students and 5 female students, which is similar to the gender distribution of the wider group. The students had a median age of 19 years, with a range of 18 to 20 years. The ethnic composition of the sample was 100% Caucasian. The analysis of social-educational indicators indicated the prevalence of students coming from families with a high educational level. All ethics issues were cautioned and all students signed informed consent statements.

### 3.2. Measurement

The chosen methods for data collection were two-fold: (i) semi-structured interview, and (ii) document analysis. Regarding the semi-structured interview, the literature advocates that this is a privileged way to capture and understand the richness, complexity, and meanings of students' choice process [25–28]. This kind of interview offered topics and questions to the interviewee, but they were carefully designed to elicit the interviewee's ideas and opinions on the topic of interest, as opposed to leading the interviewee towards preconceived choices. All the interviews were conducted, transcribed, and analysed in Portuguese.

The other method used was documental analysis, namely focusing on the institutional Guide to Strategic Development and the institutional strategic plan.

### 3.3. Data Analysis

Data analysis was performed through content analysis, using the Nvivo software. The choice of this tool was based on its versatility and flexibility to encompass the methodological orientation adopted for this research. Moreover, this software, while not being a statistical tool, took on a critical role in the efficiency of the data treatment, favouring the management and comparison of a considerable amount of non-structured data.

Transcripts were coded according to themes and analysed using a constant comparison approach [29]. The data were coded by paragraph and sentence, as proposed by Strauss and Corbin [30]. Data reduction in qualitative research is a necessary task and portions of transcripts have been selected to illustrate the respondents' views. Participants' own categories were tabulated, as suggested by Silverman [31].

## 4. Results

### 4.1. Physical Dimension

As previously stated, the physical dimension concerns the set of infrastructures composed of buildings, as well as their external and internal features. The analysis of the results from this study reveals that the main concern pointed out by the first-year students surveyed regarding this dimension regards classrooms conditions. In the newcomers' words, the comfort and effectiveness of work devices in classrooms are particularly criticised. Daniel (The names used throughout this paper are fictitious) complains that "some classrooms do not have good conditions: the chairs and tables are damaged and too small for the number of students," while Sophie goes even further in her negative assessment: "the classrooms are horrible. They are small and ugly. Only with a lot of goodwill can we have the minimum conditions." Moreover, while some students only criticise, others offer some proposals for change. Simon suggests that "classrooms, which are rather stuffy, should be ventilated. And the air conditioning, which is always broken, should be fixed." In turn, Michael presents broader suggestions, such as "heating, improved and functional structures and good lighting."

However, the criticisms and suggestions regarding the physical space are not restricted to classrooms. As mentioned above, when students start this new stage of their academic life, one of the biggest impacts felt seems to have been a confrontation with a strange physical environment. Lloyd states: "I was a bit disappointed with the picture of it." Fabien also reflects on the impact of the external look on the institutional image: "I did not know that this was so old, almost falling apart . . . It looks inconsistent since this is a Faculty of Engineering." In this discourse, the idea that the physical image of the institution should be related to its real purpose and social repercussion is evident.

When asked about the architectural changes they would propose for their new School, all first-year students interviewed suggested some kind of intervention in the physical space, to promote not only the learning, but also the well-being of the entire academic community. Thus, students interviewed suggest significant infrastructural changes. Daniel is a refurbishing supporter: "I would make an overall renovation; for example, I would improve rooms, chairs, tables. I would have the facade of the main building painted." Diana agrees: "I think the main building is very beautiful. But it would need some good works."

The importance of the physical dimension to the students' well-being is so striking that some mention it spontaneously, as a way to measure their integration. Daniel and Bruce (respectively) justify their feeling of belonging through the skills attained in terms of controlling the spatial dimension: "I don't get lost any more. I know where things are" and "I now know the school well, the places." Displaced students further extend the physical influence to their feeling of integration, alluding to the city as a catalyst macro-structure for their well-being. The words of Mark about the city hosting his School are not nice: "The city is dark and very big. It's ugly; I wouldn't like to live here forever." The comparison proposed by Nelson is not the most flattering, either: "This food is like the city, tasteless and monotonous."

The lack of green areas and recreation spaces are also criticised by the interviewees ("having more green spaces, getting spaces where students can have fun, because there are people who do not like noise or do not like smoking . . . ," Andrew). John calls for a new conception of space: "This is nice, but I would create larger and airy spaces," while Mitch and Alexis suggest the construction of a gymnasium, where they could develop sports activities ("I would build a gym. I was disappointed when I knew I could not play sports here," Mitch). Thus, the words of the interviewees are clear about the relationship between

the institutional physical dimension and the subjective dimension of well-being. Others, however, are more pessimistic about the feasibility of improving the working conditions: "I would pull it down and build it again" (James) and "It's all very old. I would build it from scratch" (Edward).

The constraints inherent in the freshmen's socio-cultural and socio-economic background are not unrelated to the conceptualisation of these approaches. Furthermore, there is a much more critical stance on the part of children of middle-class families, compared with students from other social classes. David hopes that this soon will change for the better: "I think that something could be done, but it seems that we will move soon. I hope for the better. This place is hopeless." Charles agrees: "The School is very small and is outdated in architectural terms. I hope the situation will improve with the new School . . . " Fabien reflects on the urgent need to move to new facilities: "We needed to move to a more decent place." The transition to new premises will actually occur, but it does not provide, per se, guarantees of significant improvements in academic well-being. As Tavares (2003) advocates, the changes will have to ensure better basic and specific learning experiences, to encourage the acquisition of advanced communication tools that enable students to achieve as successfully as possible the different courses, the development of a feeling of autonomy, and the development of personal and interpersonal skills to ease their social and work integration.

In fact, many studies have revealed a significant relationship between quality of physical infrastructure and student achievement [32]. For students to learn to their full potential, scientific evidence suggests that the classroom environment must be of minimum structural quality and contain cues signaling that all students are valued learners. Indoor Environmental Quality is a popular theme in all sustainable development assessment tools aimed at increasing the comfort, health, and safety of a building's occupant and their most common indicators are: thermal, acoustic and noise comfort; ventilation and contamination level; and illumination and lighting [33]. According to Barrett et al. [34], thermal comfort is related to the learning progress, i.e., students usually perform better in the classroom where the temperature is easy to control. Ergonomic comfort is also an important factor and is concerned with the study of the adaptation of a man to the work, involving the physical environment and organizational aspects related to the activities performed on site. School furniture design demonstrates a close link between school desks, health problems, and discipline in class [34]. Nonetheless, a plethora of scientific evidence suggests that student learning and achievement is deeply affected by the environment in which this learning occurs. Improving student learning, achievement, and motivation requires attending to both the structural and symbolic features in the classroom.

*4.2. Psychological and Sociological Dimensions*

Considering that individuals and their relations are the key players of the school itself, the psychological and sociological dimensions are unavoidable in its analysis. The words of Mark reflect good prospects for the psychological dimension of the Faculty of Engineering: "It's good to be in the School of Engineering!" Students' narratives suggest that the self-concept of new students is significantly increased by the feeling that they are part of a reference group, which is the explicit goal of the institutional welcome events.

There is even the assumption that the interview itself might have worked as a mode of intervention to promote academic well-being. Adopting an eminently constructivist approach, it seems that pure and simple questioning of the students' feelings and thoughts about their way of life at the school may bring about benefits, not only from the fact that students feel heard, but also the opportunity for reflecting (and sometimes expressing) on some aspects of their daily lives. Mark asserts: "I've never bothered to think much about it, but I think the lessons are good, they train well." The perception of real utility was confirmed by some of the interviewees, who asked whether these interviews would continue the following years and whether it is a common process to ask students for suggestions.

The content analysis carried out on the narratives allows us to note the ubiquity of the interpersonal relationships and their crucial importance for the individual well-being. Alexis states: "*Here the mood is great, people are fantastic.*" The feeling of integration seems to depend essentially on the relationships students establish with their peers. Andrew is explicit. "*I have colleagues whom I like, and this makes my integration.*" George also emphasises the importance of building relationships for his well-being: "*But when I have friends here, I think I'll feel better.*" This assertion is especially true when it comes to displaced students, to whom the establishment of new relationships emerges as a pressing need: "*My home community here is big. They give great help. They've been awesome, since the first day*" (Simon). Moreover, one of the big fears regarding the new university life is precisely building relationships. David, worried about facing the new relational configuration, is clear: "*What I feared most was the colleagues with whom I would share the house. Not here, we are all equal, it's different.*"

Social support is in fact important for students' first-year academic achievement. Multiple studies found that students with better quality relationships with parents, faculty members, fellow students, and high school best friends had higher GPAs in the first year [35–37]. In addition, Goguen et al. [36] found that students who had conflicts with their best university friend achieved less in the first year, while conflicts with their best high school friend did not have a significant effect. After all, as Lloyd refers ("*I like being around here because I have my friends here*"), environmental comfort depends on the quality of the interactions developed.

*4.3. Organisational Dimension*

Although these are first-year students, they too recognise that the strategic objective of any HEI should be people-centred. Michael would like, precisely, for the Board of the Engineering School to know (and acknowledge) the point of view of its students: "*Being closer to students and know their problems, their versions.*" This was one of the changes suggested by the interviewees in response to the request for proposals aimed at changing the school's community. However, if very few students interviewed stated that they would change nothing in the management of their School, the majority state their ignorance regarding the governing bodies: "*I don't know. I cannot comment on what I'm not familiar with*" (Michael). It is again Michael who reflects on the losses on this situation: "*I don't know the work of the Board. That's bad . . .*" Accordingly, Andrew even suggests: "*I don't know them. That is, I'd change that. They should make themselves known.*"

David, apparently, already knows the members of the Board and even advises them to "*be less 'full of themselves' because they feel the best.*" Michael's initial idea, which proposed a closer relationship between the governing bodies and the students, is often referred to by other colleagues. Seth thinks they should "*intervene more actively in issues of tuition, being on the students' side.*" Daniel takes the opportunity to criticise one of the measures of the Board he disagreed with: "*I think the Director was wrong by posting a notice showing up against the student demonstrations. All people have the right to express themselves.*" In this regard, students from the middle-class take on a much more critical stance, while students from the upper-class choose to express their ignorance regarding the Board and its activities. The students from the lower classes tend to be especially vocal, mainly about a closer relationship between governing bodies and students, in addition to greater availability of technical material in practical classes. The trend maintains: the importance of relationships to the lower classes, fierce criticism of the middle-classes, and a distant silence from the upper classes.

Despite these different standpoints, most proposals offered by students point towards two main mottos. On the one hand, they signal the need to increase the availability of technical and technological equipment in the classroom ("*In the electronics field there should be more material,*" Nick). On the other hand, there is the proposal for more direct intervention in the teaching strategy adopted ("*More attention to how classes are taught,*" Liam).

Taking up the respondents' words, the proposals in the pedagogical field put forth by them can be categorised into: (i) strategic pedagogical measures ("*I'd reformulate the evaluation period,*" Fabien; "*I'd increase practical classes,*" Alexis), and (ii) intervention with the faculty ("*I would make the salaries of teachers depending on student evaluations. I think it would be a good idea. Things would work differently,*" Sophie).

Regarding proposals within the scope of the more administrative functions, there are many different answers, which encompass most of the services and facilities of an essentially bureaucratic and logistical support nature provided by the school. The cafeteria seems to be the focus of major criticism, which extends not only to the diversity ("*Food should be more diversified,*" Andrew) but also the quality of food ("*Food. It's disgusting. It should be improved,*" Mark). The protagonists of these critiques are essentially displaced students and/or from lower classes, perhaps, therefore, more subject to the regular use of the canteen. The friendliness of the staff seems to be another target point of possible changes. While Anne points out that the difference in treatment that she feels concerning faculty and student populations ("*I'd make them nicer to the students . . . As for teachers, they are very pleasant*"), Sophie seems resigned to the situation: "*Sometimes, they could be nicer, but that's the way it is . . . *" Concerning the service itself, the slowness seems to be the most mentioned topic ("*fewer queues,*" James suggests), although Daniel also adds a positive critique: "*They are slow but efficient.*" Another suggestion shared by several freshmen has to do with the strategy of disseminating information, which, in their words, seems not to be the most effective: "*More logic for the signs display*" (Joseph).

Finally, a highlight ought to be made to the criticism, yet often repeated, about the bureaucracy surplus they face in their first year as HE students: "*The paperwork: they are a nuisance*" (Peter) and "*It's all very bureaucratic. There are still many obstacles*" (Liam).

### 4.4. Cultural and Political Dimension

In the specific case of the institutional culture underlying the institution under analysis, attention should be placed on its plea for the quality and recognition that it holds, as a leading HEI in the area of Engineering. It is that institutional culture, which extends to all elements of the educational community, that seems to be indisputably acknowledged and accepted by the majority, made clear by the freshmen's words: "*It is being part of an elite of the best*" (Joshua); "*It's to be VIP*" (Guy). This idea of superiority that is recognised in comparison with the other HE students, and with society in general, seems to have been fairly well integrated by the new students, who, only a few months after having joined the Faculty of Engineering, state that they feel integrated into a group with which they not only identify but also to which they are proud to belong. Their words leave no room for doubt as to the position of superiority: "*Being a student at this school is to be a bit superior to others*" (Andrew), "*Being a student of this school is to be a student of an institution of prestige earned by its requirement. It's not for all*" (Anne). Similarly, their speeches confirm the pride of belonging to this institution: "*I am very proud because of the prestige of this university*" (Nick); "*It's to have the privilege of knowing the most renowned teachers in the country and even some from abroad, and of learning from them as much as possible*" (Fabien); "*Being a student here is a privilege, because it is one of the best Engineering Schools*" (Diana). Mark even acknowledges behavioural and attitudinal idiosyncrasies to those who attend or have attended this specific school: "*There is a way of doing things peculiar of his school—a more relaxed one because we are the best.*"

It is interesting to note that it was specifically the prestige of this higher school that seems to have been an essential factor to the vocational (and strategic) option of the Bourdieu' "heirs." However, six months after their enrolment, it is mainly the first-generation students that appeal to the prestige of this Faculty, predicting a spread of that prestige for themselves as members of that institution. The identity as a HE student seems to become diluted in the face of the more specific status of an Engineering student, and the future professional status significantly influences their attitudes to the reference group: "*Being an engineer is a sign of pride, is a sign that you understand what you do*" (Mitch); "*Being an engineer is something spectacular. Not having to explain: is a kind of pride*" (Alexis). Andrew's words

are also significant: "*The place also forces us to stand for the Faculty of Engineering, we must be united.*"

However, the whole widespread wave of pride for the prestige of the School they belong to did not convince all respondents and not even all first-generation students. Nick, while acknowledging the prestige of the Faculty of Engineering, has with it a relationship based on a pragmatic and utilitarian principle: "*For me, this Faculty is a means and not an objective. I take from it what interests me: the knowledge and the degree. The rest is scenery.*" In turn, David, making use of irony that characterises much of his speech, reflected as follows: "*Now it would be nice to say that it is quite proud and stuff, right? I don't go around jumping, nor am I here forced. It's like going to the restaurant, looking at the menu and choosing the most expensive dish, as people say it is very good. Only when it comes to the table you know what it is. And only after you eat it you know it was really good. This school was the dish I chose, now we will see . . .*"

### 4.5. Axiological Dimension

To explore the axiological dimension, the Strategic Plan issued by the University, which this Faculty is part of, was analyzed. This document shows visible concerns with holistic and axiological education, inasmuch as that it defines its mission as follows: "to create scientific, cultural and artistic knowledge, high quality education strongly anchored in research, social and economic value of knowledge and the active participation in the advancement of the communities around it. The University [ . . . ] is an education, research and development institution committed to the full training of citizens, respecting their rights and actively involved in the progress of their communities" (University Strategic Plan, 2011, p. 4). In this same document, the aim of educating for the values is further strengthened by considering the main components of its mission: "The University [ . . . ] is today a national cultural, artistic, technological and scientific reference, and known also for the production and dissemination of knowledge. The University [ . . . ] is, therefore, a mobiliser and a driver of the socioeconomic and cultural development of the country" [38].

With regard specifically to the School of Engineering, it may be observed that, in its strategic plan, the institutional mission is embodied in the training of world-class engineering professionals, supported by excellent research and development, addressing the scientific, technical, ethical, and cultural aspects. It continues, stressing that, "in addition to a solid technical and scientific training, the School of Engineering will seek to give them a set of competencies and values, especially the capacity for initiative, learning and problem solving, as well as intellectual integrity and sense of responsibility and solidarity, preparing them for professional success" [39].

If the discursive intentions, whether institutional or political, seem united in the defence of the axiological component as the foundation for HE, students themselves—and particularly the freshman respondents—also seem aware of their fundamental importance. Considering the words of George on the hopes that his father lays on him, and which clearly take on the defence of values such as respect and responsibility: "*He always says: 'you'll be one of those engineers that know what they do, not like some I know, that have a lot of theory in their minds, but do not lay their hands to anything.' He also says that I have to respect people who are long in the profession, even if they do not hold a degree, because they have the degree of life*." It is the same value of respect, along with honesty, that is mentioned by George to describe the main features of a successful engineer: "*It's about understanding what you do, being supportive of people with whom you work and not being the kind 'hand over the money.'*" Among the students who appeal to moral and ethical values as key factors to professional competence, it is worth noting that the vast majority come from the middle, lower-middle, and low classes. It could be hypothesised that these social backgrounds are those which would be more aware of these values and, therefore, would instil them on their children, insofar that they possibly would have felt disadvantaged by their absence, in their own work.

On the other hand, indifference or even opposition to values taken as universally accepted, such as justice or respect for others, are pointed out as negative symptoms of

the systemic organisation in which they operate. Anne points to the peer competition as a situation to be avoided: *"In other* [Schools], *such as Medicine, it is a competition that kills them."* Hazing is, for some, also a paradigmatic example of indifference to the basic values of interpersonal relationships. Sean states: *"I felt humiliated, mistreated . . . I was not used to being treated like that."* For him, the major objective of hazing complies with the incorrect values: *"I think they are avenging of what others did to them, but we aren't to be blamed. It is a kind of revenge."*

## 5. Discussion and Conclusions

The aim of the present paper was the analysis of the HE institutional reality with a focus on the impacts of physical and cultural dimensions in the first-year students' sense of belonging. There was an interest in focusing on the diverse institutional dimensions that may act as catalysts (positive, thus facilitating, or negative, thus hindering) of the integration of new students, namely in their institutional commitment construction process. Thus, the physical, psychological, social, organisational, political and cultural, and axiological dimensions emerged as a relevant framework for analysis. This paper explored each of these dimensions of institutional analysis, using them as reading grid of the reality perceived by freshmen. All these dimensions are unavoidable as real levers for the freshmen integration process, within the institutional academic and social environment.

The physical dimension takes on great importance, insofar that it functions as a way to measure students' integration, fostering their feeling of belonging through the skills attained in terms of controlling the spatial dimension. Moreover, specific groups of students seem to be more sensitive to the physical dimension than others. On the one hand, displaced students extend the physical influence on their feeling of integration, describing the city as a catalyst macro-structure for their well-being. On the other hand, children of middle-class families seem to be more critical to a material environment that they expect to work as a lever for upward social mobility. In fact, the physical dimension of the host institution seems to work as a real catalyst for the newly arrived students' sense of belonging. The magnificence of the buildings, the quality and quality of pedagogical resources, the well-being of the classrooms, and even their decoration seem to have, in the perception of the interviewed students, a significant impact on the way they are proud to belong there. The literature in the field of education is clear about the pedagogical advantages of schools that take care of the health, safety, and comfort of their students, considered basic principles for a quality education. The spatial configuration aggregates or separates, promotes concentration or conviviality. More than functional schools, the governance of educational institutions must be attentive and promote safe, comfortable spaces with the necessary conditions of luminosity, temperature, and cosiness so that the student feels comfortable and chooses to stay there longer. Furthermore, the external signs of greatness, grandeur, and even seniority contribute to the institutional reputation and, consequently, to the feeling of pride in belonging. In the specific case of higher education, this effect of the physical/spatial dimension can be intended precisely to act as a positive catalyst for students' sense of belonging, endorsing meaningly to their commitment to the institution.

Concerning the psychological dimension, new students' self-concept emerges significantly increased by the feeling that they are part of, more than a reference group, an elite. Furthermore, regarding the sociological dimension, interpersonal relationships are crucial for the students' well-being, given that the feeling of integration seems to rely basically on the relationships they establish with their peers.

From the organisational viewpoint, students recognise that the strategic objective of any HEI should be people centred. Students from the lower classes tend to be especially demanding, mainly about a closer relationship between the governing bodies and students, while the upper-class students choose to express their ignorance regarding the Board and its activities. It is possible to identify a trend: the importance of relationships to the lower-classes, fierce criticism of the middle-classes, and a distant silence from the upper-classes'

students. Some of them do not question any factors that might in any way overshadow the prestige that was the basis of their vocational options. Others, apparently more demanding, strongly criticise an institution from which they expect the possibility of being the bridge to a higher lifestyle than their parents'. Others are concerned, above all, with the interpersonal integration as a catalyst for their institutional integration.

Moreover, in the organisational dimension, strategic pedagogical changes emerge with critical relevance, namely related to the need to envisage students as institutional protagonists. Regarding the more administrative functions, most of the services are a target of criticism, with special emphasis on the bureaucratic and logistical support.

In terms of the cultural and political dimension, the prestige that brands the HEI under analysis is recognised and accepted by all elements of the educational community. This idea of superiority seems to have been well incorporated by new students, who not only identify with the reference group, but also are proud to belong to it. This perceived prestige represented, *a anteriori*, a crucial vocational (and strategic) demand for the Bourdieu' "heirs," and, *a posteriori*, a conquest to the first-generation students, who foresee a spread of that prestige for themselves as members of that institution. In fact, the institutional culture, which in this case intentionally contributed to the creation of a feeling of upward social mobility, seems to have an effective effect on creating a sense of belonging and commitment to the institution. For students who come from families of high sociocultural status, entering this school is yet another evidence of their personal and social value, which confirms and reaffirms their social status. For first-generation students in higher education, entering this school represents the social climb not only of the student himself, but also of the family of origin, which, by cultural contagion, also ends up developing a feeling of upward social mobility.

To explore the axiological dimension, the HEI analysed shows visible concerns with the holistic and axiological education, with its aim to educate for the values in a framework of human, cultural, scientific, ethical, and technical training, in the context of diversified processes of teaching and learning and complementary activities aimed at the development of attitudes and skills, as well as the dissemination of knowledge. This axiological component is embodied and valued by students themselves, and particularly by freshmen from the middle, lower-middle, and low classes.

After analysing the students' narratives and institutional documents, a set of institutional initiatives aimed at the promotion of mechanisms to facilitate better integration of the new students emerge as significant. However, some issues that relate to the "unspoken," underlying the official nature of the purpose of the integration activities, ought to be addressed.

The massification of HE is real and tangible. One of the most obvious consequences is, undoubtedly, the emergence of new and diverse publics. The HEI analysed did, naturally, accept among its many years of history young people who, a decade or two ago, would have no viability to pursue a degree. The socio-economic and socio-cultural dispersion of the sample from this study proves that. New dimensions for the devices to manage the dichotomy between the real juvenile dispersion and the students' dispersion expected by this School are, therefore, necessary. The management of students' expectations in the face of students' realities emerges as a critical need.

If new students are not the expected ones, at least let us try to make them whom they are expected to be. This could be an underlying theme to moments such as the welcome ceremony for new students, which officially receives the newcomers and emphasises the institutional prestige over the centuries. It is explicitly conveyed to freshmen that their enrolment in this institution makes them part of an elite. If the heterogeneity characterises the population that enrols for the first time in the School of Engineering, relevant features of the attempts to homogenise those who attend it may be found, through a whole effort to build an identity framework that aims at creating feelings of belonging, pride, and group cohesion. All the ceremonies associated with the welcome of its new students, including a speech that appeals to the most distinguished alumni, seems an obvious attempt on the part

of the School of Engineering to ensure an elitist position against other HEIs, weaving, from the outset, a web, with corporatism (both in its many positive and also negative aspects) certainly as one of the main features.

However, the welcome ceremony takes just one side of a multifaceted strategy of the School of Engineering towards maintaining a place overlooking the rest of the educational landscape. The expectations risen on new students, regarding the prestige and excellence of the Faculty of Engineering, are prior to their enrolment, which is just a confirmation that verges on the Pygmalion effect. The expectations instilled especially by parents and high school teachers (many of whom are alumni of the Faculty of Engineering) create, from the outset, a golden structure that will shape the way they view and analyse the HEI at the beginning of their university path. The "snowball effect" seems to play an important role here, to which the actual quality of education provided by the institution is certainly not unrelated to. Marketing may be significant, but the "product" itself also deserves a credit reference.

The results of these strategies seem obvious by their effectiveness, in the words of the respondents: "Being a student at the School of Engineering is being a student of an institution of prestige earned by its high level of demand" (Anne).

It does not fit the purposes of this article to provide a complete answer regarding the issue that attending a degree with the seal of the School of Engineering may turn its students into an elite. In fact, it is important to recognize the limitations of this study, since, as it is a case study, it focuses on a limited number of students, on a specific institution, and on a particular program. We assume that qualitative research is an exploratory methodology. Its focus is on the subjective character of the analysed object, trying to understand the student's behaviour and studying their particularities and individual experiences, among other aspects. In addition to understanding and interpreting behaviours and trends, this methodology sought to identify hypotheses for a problem and discover the perceptions and expectations of its actors. Thus, it would be important to extend this exploration to more students, namely from other schools with cultural and physical dimensions different from those studied, in order to assess the results found here. However, it seems indisputable that the self-concept of new students is significantly increased by the feeling that they are part of a reference group, and if the psychological, sociological, and even axiological dimension are important to construct a sense of belonging to the institution, the physical and cultural dimension play a critical role as positive catalyst for the commitment that the student establishes with the institution that welcomes him. The issue is whether this recently restructured self-concept is strong enough to manage possible academic or relational failures ("*This is not an ordinary school. I came here because I deserved. It is a responsibility,*" Fabien).

Thus, if the institution does not have full control over the mechanisms that shape the profile of their incoming student population, it must, then, look after their output profile.

**Funding:** This research received no external funding.

**Institutional Review Board Statement:** The study was conducted in accordance with the Declaration of Helsinki, and approved by the Ethics Committee of Universidade Europeia (date of approval: 31 July 2020).

**Informed Consent Statement:** Informed consent was obtained from all subjects involved in the study.

**Conflicts of Interest:** The authors declare no conflict of interest.

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
