# Peer review of "The Higher Education Commitment Challenge: Impacts of Physical and Cultural Dimensions in the First-Year Students’ Sense of Belonging"

_education, doi:10.3390/educsci12040231_

Round 1
Reviewer 1 Report
The Higher Education Commitment Challenge: Impacts of Physical and Cultural Dimensions in the First-Year Students’ Sense of Belonging
- It is not clear which part of the study lines 24-45 belong to (Introduction, Materials and Methods ???).
- It is unclear why the authors of this article follow the Tavares model.
- It is also not clear why the authors resort to the concept of Environmental Psychology (63) and what role it plays in the study.
- It is also unclear what task the introduction performs in this study. The introduction is rather chaotic, replete with unacceptably large quotations from the works of Tavares.
- The explanation of the abbreviation HE is not given .
- There is no clarification of the law (136).
- A small number of respondents may question the results of the study.
- The results of the study included replicas of respondents with names. Did the authors of the study get the consent of the respondents? There is no reason to specify the names of the study participants.
- The results are not confirmed by anything. (258-273), (275-288).
- This work is not a serious scientific study, since there is no serious evidence base and its analysis, there is no literature review on this topic, there are no graphs and diagrams, diagrams and tables, the list of references is presented in a limited volume.
- This work is not a complete scientific study, but rather represents a draft version or a project for future research.

Author Response
We are re-submitting our paper " The Higher Education Commitment Challenge: Impacts of Physical and Cultural Dimensions in the First-Year Students’ Sense of Belonging". Overall, we agree with the comments of the referees. We believe that the revised version of our paper addresses all concerns by the referees in detail. As such, we believe that this new version is suitable for publication. We made sure to comply with the journal manuscript checklist and every change to the manuscript has been clearly documented below. We made some notable changes to the manuscript, which were directly requested by the referees.
We made several changes in the paper structure, revising the introductory text and the introduction, as well as the sections on materials and methods.
All literature review was updated and Tavares model was contextualized. All references to Environmental Psychology were deleted.
All citations and quotations were reviewed, as well as all abbreviation are identified.
The sample strategy was clarified.
It is important to clarify that all name used used throughout this paper are fictitious, as is explicitly stated in the disclaimer on page 7.
We also consider it important to clarify that this is a qualitative study, so its results are based on the perceptions of the individuals involved, the conflicts observed in the field and the subjective aspects identified. Since it is a case study, it is a type of research that seeks to analyze a specific situation, in a deep and complete way, in this case the o. process of integration of first year students.
In a case study, as in this paper, the researcher seeks to fully understand the object, interpreting the context in which it is inserted and the variables that influence it. The research sources for a case study were desk research and interviews.
Lastly, we thank the referees for their time spent carefully reviewing the manuscript, and in their opinions regarding the science and presentation of the material.
Your criticisms were very important for us to substantially improve the quality and clarity of the paper. So, we appreciate you helping us to improve our work.
Reviewer 2 Report
There is clearly some very interesting data here, and the paper has some potential to contribute to the literature, particularly around the relationship between status, physical location, and organisational process. That is to say, that expectations of HE status are not always as well aligned with physical location or organisation. Evidently, these things can be particularly prescient as students make the transition into higher education. Unfortunately, the current structure of the paper constrains that contribution because it rather slavishly forces a very particular model on both the literature review and the results. I would seek to make a number of points that need attention before publication. 1) This isn't really a literature review. It's an overview of Tavares' model. Fair enough perhaps, but there is no attempt made to discuss/justify these ideas, and why they are important in this particular instance. There is a very brief mention of Tinto - which is a central reference point in papers of this type - but that discussion is not sustained. 2) Indeed, none of this overview leads to a discernable 'knowledge gap' and there are no aims and objectives given for the study. Hence, it reads more like a descriptive evaluation of the engineering department. 3) The phrasing is a little awkward, particularly in the introduction, and needs some attention. 4) Details of the documentary methods used in the study are very sparse. The paper does allude to 'content analysis' but the particular type is not specified, and the later references are more commonly associated with grounded theory. Some further clarification is needed. 5) In what language were the interviews conducted/transcribed/analysed? There also seems to be some suggestion in the results that the socio-economic circumstances of the interviewees is important, but there is no information about how this was assessed in the methods. There is also no information about ethics. 6) Using Tavares' model to structure the findings constrains the development of what is interesting here - and that is the contradictions experienced between status, physical environment, and organisation. The sections on psychological and sociological dimensions do not really add anything particularly novel to the argument, and the section on 'axiological dimension' is a muddled description of policy documents, and then something on values and hazing. It is not clear how these things relate to each other, or how they are particularly relevant to the other results. 7) The discussion is primarily a re-statement of the results, and the use of Bourdieu is limited to a couple of lines. This means that the implications of the results are not assessed. Similarly, there is no attempt to examine the limitations of the study, and what that means in relation to generalisation. All of which constrains the capacity of the paper to make an original contribution. This could be an interesting case study, but it needs to be better informed by the wider literature.
Author Response
We are re-submitting our paper " The Higher Education Commitment Challenge: Impacts of Physical and Cultural Dimensions in the First-Year Students’ Sense of Belonging". Overall, we agree with the comments of the referees. We believe that the revised version of our paper addresses all concerns by the referees in detail. As such, we believe that this new version is suitable for publication. We made sure to comply with the journal manuscript checklist and every change to the manuscript has been clearly documented below. We made some notable changes to the manuscript, which were directly requested by the referees.
All literature review was updated and Tavares model was contextualized, in answer to the points 1 and 2 of your review report.
We made several changes in the paper structure, revising the introductory text and the introduction, as well as the sections on materials and methods, in answer to the points 3, 4 and 5 of your review report.
we tried to overcome the limitations identified in relation to the analysis carried out in the psychological, sociological and axiological dimensions. We hope to have managed to make these analyzes more relevant.
in the results section, we tried to make the discussion more in-depth and related to the literature review.
We add reference to the limitations of this study.
Lastly, we thank the referees for their time spent carefully reviewing the manuscript, and in their opinions regarding the science and presentation of the material. Your criticisms, suggestions and comments were very important for us to substantially improve the quality and clarity of the paper. So, we appreciate you helping us to improve our work. We are also grateful for the cordial and friendly manner in which the criticisms were made. :)
Round 2
Reviewer 1 Report
I thank the authors for the substantial revision of the material, which improved the study.
Reviewer 2 Report
I'm happy with the changes made to the paper. It is now clearly grounded in relevant literature.